# *SCN1A* Variants as the Underlying Cause of Genetic Epilepsy with Febrile Seizures Plus in Two Multi-Generational Colombian Families

**DOI:** 10.3390/genes13050754

**Published:** 2022-04-25

**Authors:** Diana M. Cornejo-Sanchez, Anushree Acharya, Thashi Bharadwaj, Lizeth Marin-Gomez, Pilar Pereira-Gomez, Liz M. Nouel-Saied, Deborah A. Nickerson, Michael J. Bamshad, Heather C. Mefford, Isabelle Schrauwen, Jaime Carrizosa-Moog, William Cornejo-Ochoa, Nicolas Pineda-Trujillo, Suzanne M. Leal

**Affiliations:** 1Center for Statistical Genetics, Gertrude H. Sergievsky Center, and the Department of Neurology, Columbia University Medical Center, New York, NY 10032, USA; dmc2245@cumc.columbia.edu (D.M.C.-S.); aa4471@cumc.columbia.edu (A.A.); tb2890@cumc.columbia.edu (T.B.); liznouel1308@gmail.com (L.M.N.-S.); is2632@cumc.columbia.edu (I.S.); 2Gene Mapping Group, Faculty of Medicine, University of Antioquia, Medellin 050010470, Colombia; lizethmaring@hotmail.com (L.M.-G.); pilipereira17@gmail.com (P.P.-G.); jaime.carrizosa@udea.edu.co (J.C.-M.); 3Department of Genome Sciences, University of Washington, Seattle, WA 98195, USA; codavis@uw.edu (UWCMG); mbamshad@uw.edu (M.J.B.); heather.mefford@stjude.org (H.C.M.); 4Department of Pediatrics, University of Washington, Seattle, WA 98105, USA; 5Pediatrics Group, Faculty of Medicine, University of Antioquia, Medellin 050010470, Colombia; wcornejochoa@gmail.com; 6Taub Institute for Alzheimer’s Disease and the Aging Brain, Columbia University Medical Center, New York, NY 10032, USA

**Keywords:** epilepsy, GEFS+, *SCN1A*, incomplete penetrance, autosomal dominant

## Abstract

Genetic epilepsy with febrile seizures plus (GEFS+) is an autosomal dominant disorder with febrile or afebrile seizures that exhibits phenotypic variability. Only a few variants in *SCN1A* have been previously characterized for GEFS+, in Latin American populations where studies on the genetic and phenotypic spectrum of GEFS+ are scarce. We evaluated members in two multi-generational Colombian Paisa families whose affected members present with classic GEFS+. Exome and Sanger sequencing were used to detect the causal variants in these families. In each of these families, we identified variants in *SCN1A* causing GEFS+ with incomplete penetrance. In Family 047, we identified a heterozygous variant (c.3530C > G; p.(Pro1177Arg)) that segregates with GEFS+ in 15 affected individuals. In Family 167, we identified a previously unreported variant (c.725A > G; p.(Gln242Arg)) that segregates with the disease in a family with four affected members. Both variants are located in a cytoplasmic loop region in SCN1A and based on our findings the variants are classified as pathogenic and likely pathogenic, respectively. Our results expand the genotypic and phenotypic spectrum associated with *SCN1A* variants and will aid in improving molecular diagnostics and counseling in Latin American and other populations.

## 1. Introduction

Epilepsy is a common neurological disorder characterized by debilitating seizures and impaired cognitive function, among other etiologies [1]. It is caused by hypersynchronous electrical activity in the brain’s neuronal networks, impacting cognitive, psychological, neurological, and social aspects of life [2]. Approximately 50 million people worldwide are afflicted with some form of epilepsy [3]. The prevalence in low to middle income countries is higher, wherein ~140 per 100,000 individuals are affected compared to ~49 per 100,000 individuals in high income countries [3]. These differences may be due to disparity in health care access, cultural beliefs leading to underreported figures (e.g., in some countries where epilepsy is stigmatized), regional environmental exposures (e.g., epilepsy caused by infectious agents), and socioeconomic status [4].

Early stages of gene discovery in epilepsy involved the use of family data (e.g., Mendelian inherited epilepsies), followed by genome-wide association analysis which was unsuccessful in finding new causal genes, and more recently a period of genome sequencing and fast gene discovery (e.g., epileptic encephalopathies) [5]. These new epilepsy genes have shown the marked genetic heterogeneity and the possible pathophysiological mechanisms (e.g., ion channel disruptions, synaptic protein dysfunction, mTOR pathway, chromatin remodeling, and transcription regulators) that contribute to the etiological landscape of the epilepsies [5,6].

Genetic epilepsy with febrile seizures plus (GEFS+) is a familial, phenotypically heterogeneous syndrome with an autosomal dominant mode of inheritance and incomplete penetrance. The phenotypic spectrum ranges from mild febrile seizures (FS) and febrile seizures plus (FS+) to the devastating Dravet syndrome or severe myoclonic epilepsy of infancy (SMEI) [7]. Clinical research has shown that members GEFS+ families can present with any of the following phenotypes: (1) FS, i.e., seizures occurring with fever between three months and six years of age; (2) FS+ i.e., FS extending beyond six years of age or afebrile tonic–clonic seizure can also occur; (3) FS/FS+ with generalized (e.g., absences, myoclonic, tonic–clonic, atonic) or focal seizures; (4) afebrile generalized tonic–clonic seizures alone (AGTCS); (5) classical genetic generalized epilepsy (GGE) syndromes that include childhood absence epilepsy (CAE), juvenile absence epilepsy (JAE), juvenile myoclonic epilepsy (JME), or epilepsy with generalized tonic–clonic seizures alone (EGTCSA); (6) epilepsy with myoclonic–atonic seizures (EMA); or (7) focal epilepsy alone (FE) [8,9].

Approximately 10% of familial GEFS+ is due to variants in *SCN1A* (MIM 182389) and about 90% of these are missense [10,11,12]. *SCN1A* is a member of a ten gene family and encodes Na_v_1.1, a large α subunit of the heteromeric voltage-gated transmembrane sodium ion channel mediating neuronal and muscular excitability in vertebrates. Na_v_1.1 is a transmembrane protein with four homologous domains (DI-DIV), each with six transmembrane segments (S1–S6). *SCN1A* is highly expressed throughout the brain, with some expression also observed in the lungs and reproductive organs. So far, over 1100 pathogenic and likely pathogenic (PLP) variants for epilepsy, seizures, and hemiplegic migraines in *SCN1A* have been reported, with most of them located in the C-terminus and pore loops and the majority of variants identified in patients with SMEI. *SCN1A* is one of the most consequential genes for ion channelopathies, such as GEFS+, SMEI, and familial hemiplegic migraine [11]. Even though there are several world-wide reports of *SCN1A* variants in GEFS+ families, there is a lack of studies in Latin American families, especially from Colombia. The type and distribution of variants, phenotype–genotype correlations, phenotypic spectrum, and the presence of genetic modifiers is largely unknown in the Colombian population.

In this study, we analyzed two large multi-generational Colombian families of Paisa ancestry with GEFS+ to further our understanding of underlying variants in a population that is not widely reported. Paisa are individuals of mixed European and Amerindian ancestry from North-West Colombia, i.e., the regions of Antioquia, Caldas, Risaralda, Quindío, northern Valle del Cauca, and western Tolima. We report on two heterozygous *SCN1A* (NM_001165963.1) variants that segregate with an autosomal dominant mode of inheritance with incomplete penetrance in the two families. Family 047 is consanguineous and segregates (c.3530C > G; p.(Pro1177Arg)), and Family 167 segregates (c.725A > G;p.(Gln242Arg)) and is non-consanguineous. 

## 2. Materials and Methods

### 2.1. Standard Protocol Approvals, Registrations, and Patient Consents

The study was approved by the Institutional review board (IRB) of Columbia University Medical Center (IRB-AAAS3435) and the IRB of the Faculty of Medicine at University of Antioquia (Medellin, Colombia). All participating individuals and parents/guardians of minors provided written informed consent. Minors older than 12 years of age also provided assent. 

### 2.2. Clinical Assessment

We collected blood samples from 19 affected and 29 unaffected individuals of consanguineous Family 047 (Figure 1) and five affected and nine unaffected individuals of non-consanguineous Family 167 (Figure 2). DNA was extracted using standard procedures [13]. In addition to reviewing their seizure history and medical records, all available family members were also evaluated by a neurologist. Following clinical assessment and review of medical records and electroencephalograms (EEG), seizure types and epilepsy syndromes were classified according to the International League Against Epilepsy (ILAE) criteria [14].

### 2.3. Exome Analysis

DNA samples from individuals VI:3 and VI:40 presenting with epilepsy, VII:10 with sleep paralysis (Family 047), and individual IV:6 diagnosed with epilepsy (Family 167) underwent exome sequencing. Before exome sequencing, the selected samples were evaluated using a 63-SNP OpenArray assay derived from a custom exome SNP set to evaluate DNA quality, to confirm biological sex, and to provide a molecular fingerprint. Exome library preparation was performed using the Twist+RefSeq Human Core Exome kit (Twist Bioscience, San Francisco, CA, USA). Paired-end sequencing was performed on a NovaSeq6000 instrument (Illumina Inc., San Diego, CA, USA), with an average sequencing depth of target regions of ~31.9× (Family 167) and ~33× (Family 047). After removing low-quality reads, the filtered reads were aligned to the human reference genome (GRCh37/hg19) using Burrows–Wheeler Aligner-MEM (BWAv0.7.15) [15]. Duplicate reads were marked using Picard-tools (v2.5.0) [16]. Single nucleotide variants (SNVs) and insertion/deletion (InDel) were called InDel realignment, and base quality score recalibration was performed with Genome Analysis Toolkit (GATK) (v3.7) [17]. Additionally, CONiFER (v0.2.2) was used to call copy number variants (CNVs) [18]. 

Next, variant annotation was performed using ANNOVAR [19], including dbnsfp35a and dbscSNV1.1 databases, and for CNVs via a custom script annotating information from the BioMart Database [20] and Database of Genomic Variants [21]. SNV, InDel, and CNV filtering was performed using in-house-tools considering an autosomal dominant mode of inheritance and variants were selected for further investigation that were predicted to have an effect on protein function or pre-mRNA splicing (missense, nonsense, frameshift, start–loss, splice region, etc.) with a population-specific minor allele frequency (MAF) of <0.0005 in every population of the Genome Aggregation Database (gnomAD)(v2.1.1) [22] and Database of Genomic Variants [21]. Variants were further prioritized based on bioinformatic prediction, e.g., Combined Annotation Dependent Depletion (CADD) and Genomic Evolutionary Rate Profiling (GERP++) scores [23,24], and if they were located in human/animal epilepsy or seizure genes [25,26]. Candidate variants were visualized with the Integrative Genomics Viewer (IGV 2.4.3) [27] and their frequencies in the Bravo TOPMed database [28] were also obtained.

### 2.4. Sanger Sequencing

To validate and test the candidate variants discovered through exome sequencing for segregation, Sanger sequencing was performed using Polymerase Chain Reaction (PCR) followed by direct sequencing of the PCR product on an ABI3130XL sequencer (Applied Biosystems Inc., Foster City, CA, USA) in family members for whom DNA was available. The chromatogram of each individual was visualized and carefully scanned to determine the genotype. 

## 3. Results

### 3.1. Clinical Findings

For Family 047 (Figure 1), genetic and clinical data are available for family members in four generations (Appendix A). The largest branch of the pedigree originates from the consanguineous marriage of individuals IV:1 and IV:2 who are first cousins and have 60 descendants. Affected members of this family with a formal diagnosis have FS, FS+, or AGTCS. No individuals of this family were reported to have classical GGE, EMA, or FE. 

For Family 167 (Figure 2), the pedigree structure for five generations was constructed. Clinical and genetic data were available for family members up to three generations. Affected individuals in this family all presented with FS or FS+.

### 3.2. Exome Analysis

After analyzing exome data of VI:3, VI:40, and VII:10 of Family 047, we observed a missense variant in *SCN1A* (c.3530C > G; p.(Pro1177Arg)) present in individuals with epilepsy (VI:3 and VI:40) but not in the unaffected individual VII:10. No genes relevant to sleep paralysis were observed in this family. The *SCN1A* variant was validated using Sanger sequencing and it segregates with epilepsy with incomplete penetrance in family members with an available DNA sample. The variant is absent from gnomAD, Database of Genomic Variants, and TOPMed. The substitution was predicted damaging by several bioinformatic tools (CADD = 28.6, SIFT = 0.9, LTR = 0.6, MutationTaster = 0.8) and is located at an evolutionary conserved residue in mammals, lower vertebrates, and invertebrates (Appendix A). In ClinVar [29], it was reported as a variant of unknown significance (VUS) (Accession number: VCV000521780.6) based on two cases: one with a history of neurodevelopmental disorder (MedGen UID: 751520), and another with an inborn genetic disease and early onset Dravet syndrome (MedGen UID: 181981). Based on our findings, and in accordance with the guidelines of American College of Medical Genetics (ACMG) [30], this variant can be reclassified as pathogenic. 

Candidate variant *SCN1A* (c.725A > G; p.(Gln242Arg)) was identified in the exome data obtained from IV:6, who is a member of Family 167 which segregates with GEFS+. This variant is absent from gnomAD, Database of Genomic Variants, and TOPMed databases and is predicted to be damaging by various bioinformatic tools (CADD = 23.7, SIFT = 0.8, LTR = 0.6, MutationTaster = 0.6) and this position is evolutionarily conserved between species (Appendix A). This is also the first report of the variant. Based on our findings and in accordance with guidelines of ACMG, the p.(Gln242Arg) variant can be classified as likely pathogenic. 

No CNVs or variants in other genes implicated in GEFS+ were found to segregate in either family (Appendix A). Both p.(Pro1177Arg) and p.(Gln242Arg) variants have been deposited in ClinVar (Accession Numbers: SCV001478263 and SCV001478264).

### 3.3. Sequencing Analysis and Penetrance

Sanger sequencing of *SCN1A* variant c.3530C > G in all DNA samples available for Family 047 showed that the variant was heterozygous in all affected members and three unaffected individuals who are obligate carriers (IV:2, V:15 and IV:3) and one unaffected member (V:23), who is the son of obligate carrier IV:3 and has two affected brothers (Figure 1). The penetrance of the c.3530C > G in this family was estimated to be 81.8%. 

In Family 167, all affected members and two obligate carriers were heterozygous for variant c.725A > G in *SCN1A*. The penetrance of the variant was estimated to be 83.3%. Index case V:2 could not be tested due to DNA depletion prior to Sanger sequencing and a new sample cannot be obtained. 

## 4. Discussion

Here we report two heterozygous variants in *SCN1A,* p.(Pro1177Arg) and p.(Gln242Arg), in two Paisa families from Colombia. Both families display an autosomal dominant inheritance pattern with incomplete penetrance. *SCN1A* has been widely implicated in epilepsy with varying severity and both genetic and phenotypic heterogeneity. Previously, 44 PLP and VUS variants have been reported for GEFS+ in the *SCN1A* gene and over 86% of these lie in DI-DIV domains (Figure 3). About half of these variants are found within domains DII and DIII (n = 22). Just over 13% (n = 6) of the variants are found outside these four domains. Variant p.(Gln242Arg) localizes in the cytoplasmic loop between S4 and S5 of domain DI, and p.(Pro1177Arg) localizes to the large cytoplasmic loop between S6 of domain DII and S1 of domain DIII. Contrary to this, the bulk of variants previously associated with GEFS+ have been typically found in the transmembrane and pore forming regions (Figure 3). In general, cytoplasmic loops are less well-conserved than transmembrane domains in voltage-gated sodium channels. 

The variant in Family 047, p.Pro1177, is located in the same cytoplasmic region as p.Trp1204 and p.Thr1174. A functional variant was expected to be found at *SCN1A* gene since previous linkage analysis, using STRs, for this family had provided a logarithm of the odds (LOD) score > 6 for the febrile seizure locus *FEB3* [31]. Functional evaluation of a GEFS+ variant p.(Trp1204Arg) performed in the orthologous rat sodium channel (rNa_v_1.1) and expression in *Xenopus* oocytes have both demonstrated an alteration in voltage-dependent gating, resulting in either neuronal hypo- or hyperexcitability [32]. However, electrophysiological measurements of transfected human kidney cells with p.(Trp1204Arg) variant demonstrated a reduction in both current density and neuronal firing frequency in a heterozygous state, suggesting hypoexcitability and that the variant causes a loss-of-function (LoF) [33]. Based on the mixed effects on function, coupled with the presentation of a homogeneously milder phenotype than Dravet syndrome in all GEFS+ affected members of Family 047, we suggest an altered or hypomorphic effect in which the complete functioning of Na_v_1.1 is not completely abolished [12]. However, further electrophysiological studies are essential to fully assess the functional characteristics of this variant in the protein and its effect on epilepsies [34].

So far, there are no other variants associated with GEFS+ reported in the same functional region of DI as p.(Gln242Arg). However, based on the consistently mild phenotype presented by affected members of Family 167, we believe a hypomorphic effect is also the most likely disease mechanism. Further electrophysiological experiments would be needed to determine a precise mechanism. 

Aside from this report, there was one previous study in another large Colombian family with 13 GEFS+ individuals, which reported a missense variant (c.5213 A > G; p.(Asp174ly)) in the pore-forming region of DIV of Na_v_1.1 [10]. This indicates high allelic heterogeneity in a genetically special population for GEFS+. Additionally, phenotypic heterogeneity is observed intra- and inter-pedigrees. A more severe phenotype was observed in some of the affected individuals from the pedigree reported previously, including astatic myoclonic seizures [10].

So far, outside of Colombia, PLP *SCN1A* variants associated with seizures have only been reported in three additional Latin American countries: Argentina, Brazil, and Mexico [35,36,37,38]. Further investigation of the relatively homogenous Paisa population may aid in identification of new gene associations and better genotype–phenotype correlation of *SCN1A* variants. A comprehensive database of GEFS+ variants, their effects, and associated phenotypes to help diagnoses and genetic counseling is lacking for Latino populations. Phenotypic rescue of mice with Dravet syndrome, the more severe form of epilepsy due to *SCN1A* variants with mostly LoF variants, has led to development of several gene therapy approaches including oligonucleotides and viral vectors [39]. Therefore, further identification and functional evaluation of *SCN1A* mutations could contribute to improved diagnosis and treatment of these patients. 

In conclusion, we present here, for the first time to our knowledge, heterozygous variants in *SCN1A* in two Paisa families: a pathogenic variant (c.3530C > G; p.(Pro1177Arg)) in a consanguineous family and a likely pathogenic variant (c.725A > G; p.(Gln242Arg)) in a non-consanguineous family. This study expands the genotypic spectrum of *SCN1A* variation and will aid in improving and guiding molecular diagnosis in Latin American populations. 

## Figures and Tables

**Figure 1 genes-13-00754-f001:**
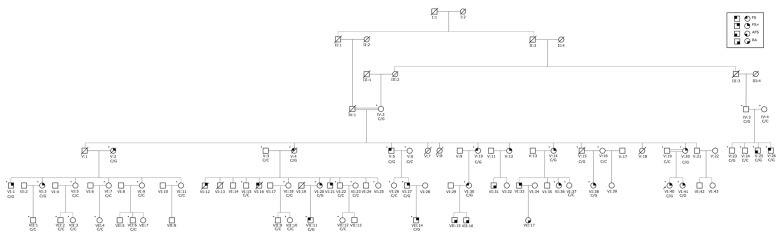
Pedigree of Family 047: Squares represent males and circles females. Clear symbols are unaffected family members. The solid quadrants represent the following diagnoses: top left solid quadrant—febrile seizures (FS); top right solid quadrant—febrile seizures plus (FS+); and bottom left solid quadrant—afebrile generalized tonic-clonic seizures (AFS). Individuals with a solid bottom right quadrant were reported to have epilepsy, but there was no available clinical evaluation made by a neurologist. All family members with an available DNA sample are marked with a ‘+’ symbol on the top left corner. A single arrow at the bottom left represents the index case. Genotypes for the pathogenic variant [c.3530C > G; p.(Pro1177Arg)] are shown below each family member for which there is an available DNA sample.

**Figure 2 genes-13-00754-f002:**
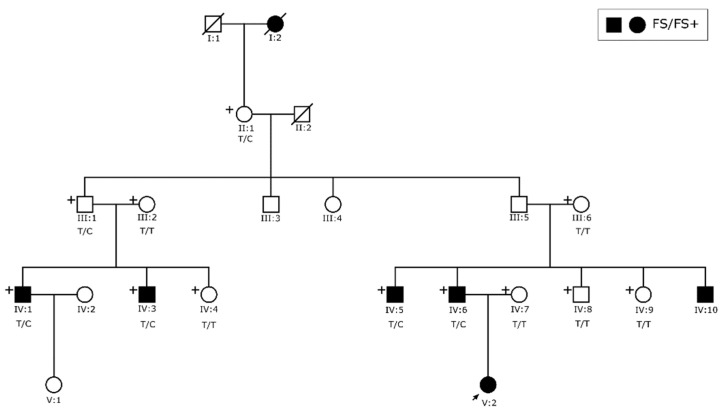
Pedigree of Family 167. Squares represent males and circles females. Solid symbols signify that the individual presents with FS/FS+ and clear symbols are unaffected family members. All members with an available DNA sample are marked with a ‘+’ symbol on the top left corner. A single arrow at the bottom left represents the index case. Genotypes for the likely pathogenic variant c.725A>G; p.(Gln242Arg) are shown below each family member for which there is an available DNA sample.

**Figure 3 genes-13-00754-f003:**
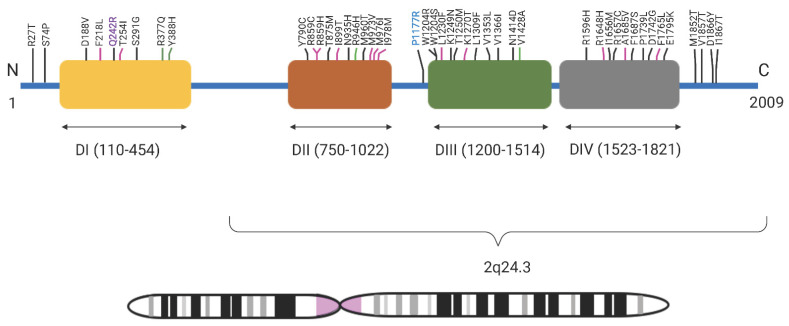
GEFS+ variants in SCN1A: All reported SCN1A (chromosome 2q24.3) GEFS+ pathogenic, likely pathogenic variants and variants of unknown significance reported in SCN1A so far showing their location in the protein in four domains—D1, DII, DIII, and DIV. Black lines indicate topological domain, pink lines indicate transmembrane domain, and green lines indicate intramembrane domain. In bold purple is the pathogenic variant p.Q242R variant reported in Family 047 and in bold blue is the likely pathogenic p.P1177R variant reported in Family 167.

## Data Availability

The data that supports the findings of this study are available in the Appendix A of this article.

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
