# Peer review of "SCN1A Variants as the Underlying Cause of Genetic Epilepsy with Febrile Seizures Plus in Two Multi-Generational Colombian Families"

_genes, 2022, doi:10.3390/genes13050754_

Round 1

Reviewer 1 Report

The authors present the article entitled “SCN1A variants as an underlying cause of genetic epilepsy with febrile seizures plus in two multi-generational Colombian families”.

I encourage the authors to read carefully my comments/recommendation according to the following concerns:

General concerns:

  • Abstract: Avoid using headings in a single paragraph. I encourage the authors to read carefully the instructions for the authors.
  • References must be numbered in order of appearance in the text and listed individually at the end of the manuscript. I encourage the authors to read carefully the instructions for the authors.
  • Check out for typos: for instance <MutationTester >(line 248).

More specific concerns:

Introduction section could be reviewed to give a more broad view of the current state of the research field and cite key publications.  I suggest to include briefly the objective/aim of study at the end of the introduction and the structure of the manuscript.

Line 39. Check the reference Liu et al. 2013 it seems not appropriate.

Section 2. Materials and Methods

Subsection 2.3 Exome analysis

  • Line 99: <affected members VI:3, VI:40, and VII:10 (family 047)>

Based on the pedigree, VII:10 is not affected and presents a wild type genotype C/C. Please clarify.

  • Line 99-100:<..,and IV:5 (Family 167)…> The sample IV:5 doe not correspond to the sample mentioned in Section 3, Subsection 3.2 line 175 <…data obtained from sample IV:6…> Please clarify
  • Line 97 cite the reference Scheffer IE, Berkovic S, Capovilla G, Connolly MB, French J, Guilhoto L, Hirsch E, Jain S, Mathern GW, Moshé SL, Nordli DR, Perucca E, Tomson T, Wiebe S, Zhang YH, Zuberi SM. ILAE classification of the epilepsies: Position paper of the ILAE Commission for Classification and Terminology. Epilepsia. 2017 Apr;58(4):512-521. doi: 10.1111/epi.13709. Epub 2017 Mar 8. PMID: 28276062; PMCID: PMC5386840.

Section 3. Results

Subsection 3.2 Exome analysis

  • Line 160: <After analyzing exome data of VI:3, VI:40, and VII:3 of Family 047>

Previously (line 99), the authors wrote: <VII:10> and  here (line 160) <VII:3>. Based on the pedigree VII:3 is also an healthy individual with WT genotype C/C. Please clarify which sample has been sequenced.

Section 3. Results

Subsection 2.2 Clinical Assessment

  • The clinical information is scarce.

I recommend the authors to provide a table with an overview over the clinical data (age; age at onset/remission; type of seizures/numbers; EEG; anti-epileptic medicaments; other neurological findings; Clinical classification of epilepsy)  which I would find helpful.

Figures:

Figure 1: Pedigree of family 047. The image is not easy to read. The family ID on the top of the pedigree is not consistent with what reported along the manuscript. The meaning of all acronyms of the legend like RA must be defined in the caption.  

 Supplementary information

  • In the “Supplementary Table of Variants Tested” the authors wrote: < Homozygous variant present only in affected member VI:41…..>

Which DNA sample was analyzed by WES ?  VI:41 or VI:40 (lines 99 and 160)? Please clarify

  • In the “Supplementary Table of Variants Tested” the authors wrote: < Heterozygous variant present only in affected member VI:2…..>

Which DNA sample was analyzed by WES ?  VI:2 or VI:3  (lines 99 and 160)? Please clarify

Conclusion:

The authors nicely reported in Figure 3 several SCN1A variants associated with GEFS+ (which might not be all according to the HGMD database and to the SCN1A database) and their schematic position respect in the protein. The authors suggest an altered or hypomorphic effect of the variant p.Thr1174 based on the evidences that the mutation p.Trp1204Arg located in the same cytoplasmic region, has been demonstrated to have such effect. This implicate that it might possible to infer, based on the location of the SCN1A missense variant, the effect on the channel activity hyperexitability or hypoexitability. Could the authors provide such evidences for GEFS+ associate mutations or comments on that? In this regard, I would suggest the authors the following paper Brunklauset al., 2020  PMID: 31782251.

The description of novel SCN1A variants associated with GEFS+ in two Colombian families is certainly interesting and the paucity of studies in Latin American population gives the work an additional value therefore I suggest the authors to highlight in a more structured way the value/novelty.

Reviewer 2 Report

The manuscript by Cornejo-Sanchez et al reports on two distinct familial SCN1A variants, identified in members of two separate multigenerational Colombian families of Paisa ancestry. Both variants show an autosomal dominant inheritance pattern with incomplete penetrance. The c.3530C>G SCN1A mutation is recurrent and the affected members of the first family studied (Family 047) present febrile seizures (FS), febrile seizure plus (FS+), or afebrile generalized-tonic clonic seizures (AGTCS). The p.Pro1177Arg mutation is localized on the large intracellular loop between Domain II-III of the voltage gated Nav1.1 channel. The c.725A>G SCN1A mutation has not previously been reported. The affected members of the second family studied (Family 167) present FS or FS+. The p.Gln242Arg mutation is localized in S4-5 of Domain I of Nav1.1.

So far there are only a few studies in Latin American families with SCN1A variants associated with GEFS+ syndrome, and information on genotype-phenotype correlations and the phenotypic severity of affected patients, especially in the Colombian population, is scarce. The results highlight that such information is valuable for diagnosis and patient counselling and could as well contribute to improved treatment of GEFS+ patients with SCN1A variants.

The methods and results are presented clearly, and the paper is well-written. There are a few concerns that should be addressed to ensure the paper has the highest possible impact. 
Abstract: the sentence “Although several variants in SCN1A have been previously characterized for several forms of epilepsy, including GEFS+” is circumstantial and should be simplified, e.g., ‘only few SCN1A variants associated with epilepsy, including GEFS+..’

Intro, line43: “The prevalence in low to middle income countries is higher” compared to high income countries – please consider including an explanatory sentence to clarify if this is due to better specialist care and access to antiseizure medications.

Figure 3 needs to be adjusted. Please add the description of chromosome 2, shown at the bottom of the figure; 2q24.3, the coordinates describing the exact location of SCN1A, should briefly be explained. Note that while p.Thr1174 and p.Thr1177 are localized on the large intracellular/cytoplasmic loop, p.W1204 is localized on S0 of Domain III (line 217).

Regarding the scores obtained with in silico/bioinformatics tools, CADD, SIFT, LTR, etc., indicate pathogenic and likely pathogenic status for the p.Pro1177Arg and p.Gln242Arg variants, respectively. However, the functional impacts of these variants on neuronal excitability is more difficult to predict - in particular for the p.Pro1177Arg localized on a channel region with unresolved structure. The assumption that that p.Pro1177Arg would exhibit functional properties similar to that of p.Trp1204Arg and lead to a hypomorphic outcome should be softened. The functional impacts of the variants need to be clarified by follow up electrophysiological studies, as suggested in line 235.

Minor

- please spell out ‘febrile seizure locus’ FEB3 in the Discussion (line 219).

- line 219: the reference (Gomez, n.d.) needs a date or to be explained. 

Round 2

Reviewer 1 Report

The authors have addressed my comments.